# Comparison and Validation of Rapid Molecular Testing Methods for Theranostic Epidermal Growth Factor Receptor Alterations in Lung Cancer: Idylla versus Digital Droplet PCR

**DOI:** 10.3390/ijms242115684

**Published:** 2023-10-27

**Authors:** Camille Léonce, Clémence Guerriau, Lara Chalabreysse, Michaël Duruisseaux, Sébastien Couraud, Marie Brevet, Pierre-Paul Bringuier, Delphine Aude Poncet

**Affiliations:** 1Department of Pathology, Tumor Molecular Biology Unit, Groupement Hospitalier Est, Hospices Civils de Lyon, 69394 Bron, France; camille.leonce@chu-lyon.fr (C.L.); clemence.guerriau@chu-lyon.fr (C.G.); lara.chalabreysse@chu-lyon.fr (L.C.); m.brevet@biwako.fr (M.B.); pierre-paul.bringuier@chu-lyon.fr (P.-P.B.); 2University of Lyon, Université Claude Bernard Lyon 1, 69100 Lyon, France; michael.duruisseaux@chu-lyon.fr (M.D.); sebastien.couraud@chu-lyon.fr (S.C.); 3Cancer Research Center of Lyon, UMR INSERM 1052 CNRS 5286, 69008 Lyon, France; 4CNRS UMR 5261, INSERM U 1315, LabEx DEVweCAN, Institut NeuroMyoGène (INMG), Pathophysiology and Genetics of the Neuron and Muscle (PGNM) Laboratory, Team Chromatin Dynamics, Nuclear Domains, Virus, 69008 Lyon, France; 5Respiratory Department and Early Phase, Louis Pradel Hospital, Hospices Civils de Lyon Cancer Institute, 69100 Lyon, France; 6Department of Pulmonology and Thoracic Oncology, Lyon Sud Hospital, 69495 Pierre Bénite, France

**Keywords:** epidermal growth factor receptor, diagnosis, Idylla, ddPCR, method validation, lung cancer, NSCLC, LUAD

## Abstract

Targeting EGFR alterations, particularly the L858R (Exon 21) mutation and Exon 19 deletion (del19), has significantly improved the survival of lung cancer patients. From now on, the issue is to shorten the time to treatment. Here, we challenge two well-known rapid strategies for *EGFR* testing: the cartridge-based platform Idylla™ (Biocartis) and a digital droplet PCR (ddPCR) approach (ID_Solution). To thoroughly investigate each testing performance, we selected a highly comprehensive cohort of 39 unique del19 (in comparison, the cbioportal contains 40 unique del19), and 9 samples bearing unique polymorphisms in exon 19. Additional L858R (N = 24), L861Q (N = 1), del19 (N = 63), and WT samples (N = 34) were used to determine clear technical and biological cutoffs. A total of 122 DNA samples extracted from formaldehyde-fixed samples was used as input. No false positive results were reported for either of the technologies, as long as careful droplet selection (ddPCR) was ensured for two polymorphisms. ddPCR demonstrated higher sensitivity in detecting unique del19 (92.3%, 36/39) compared to Idylla (67.7%, 21/31). However, considering the prevalence of del19 and L858R in the lung cancer population, the adjusted theranostic values were similar (96.51% and 95.26%, respectively). ddPCR performs better for small specimens and low tumoral content, but in other situations, Idylla is an alternative (especially if a molecular platform is absent).

## 1. Introduction

Lung cancer is the main cause of cancer-related deaths worldwide [1]. Non-small cell lung cancer (NSCLC) constitutes the major histological subtype, accounting for 85% of cases and encompassing lung adenocarcinoma (LUAD, 60%), squamous cell carcinoma (30%), and large cell carcinoma (10%) [2]. The incidence of NSCLC has been rising since the 2000s, as well as the proportion of never-smoker patients, whose tumors are frequently mutated for *EGFR* [3,4]. The development of tissue sequencing methods has improved the molecular characterization and therapeutic management of NSCLC patients through personalized treatment [5]. Indeed, epidermal growth factor receptor (*EGFR*) gene alterations are present in 11 to 30% of NSCLC cases and result in constitutive activation of the receptor [6,7]. Approximately 85% of *EGFR* mutations occur either in exon 19, involving a deletion of 15 to 18 nucleotides (del19), or in exon 21, leading to the substitution of lysine at position 858 with arginine (L858R) or glutamine at position 861 (L861Q) [8]. These alterations cause conformational changes in the EGFR’s ATP binding site, leading to abnormal activation of downstream signaling pathways responsible for oncogenic addiction in tumor cells. Tyrosine kinase inhibitors (TKIs), ATP competitive molecules targeting these pathological conformations, have been developed. In metastatic stages, first-generation TKIs (erlotinib or gefitinib) have significantly improved patient response and survival compared to standard approaches using chemotherapy [9,10]. Third-generation TKIs, such as osimertinib, also target the acquired resistance mutation *EGFR* T790M and have demonstrated a higher efficacy [11,12]. Recently, the ADAURA trial showed that using osimertinib in adjuvant settings, after surgery and adjuvant chemotherapy, improves overall survival compared to a placebo approach [13]. Thus, the identification of *EGFR*-activating mutations on tumors is crucial in clinical routine for initiating EGFR-TKI therapy. Next-generation sequencing (NGS) is commonly used to test the *EGFR* mutational status on tumor samples, but it is expensive and requires time and trained staff for both experimental procedures and result interpretation. The delay in obtaining NGS results is another important limitation, as some patients experience rapid tumor progression. Hence, a recent study involving a cohort of 96 patients revealed that 6% of patients died before obtaining NGS results, even though they could have benefited from EGFR-TKIs [14]. Therefore, for such cases, the challenge is the rapid detection of *EGFR*-activating mutations [15].

Various rapid approaches, such as qPCR with conventional platforms (Therascreen, Qiagen, Cobas), the cartridge-based fully automated system Idylla, and absolute quantification using droplet digital PCR (ddPCR), can accelerate the genotyping of *EGFR*. The Idylla cartridge, composed of microfluidics and preloaded reagents, accepts either formalin-fixed-paraffin-embedded tissue (FFPE) ribbon, extracted DNA, or biological fluids (such as fine-needle aspiration liquid [16]), and carries out deparaffinization, DNA extraction, and PCR amplification [15,17] to detect *EGFR* mutations in a quick timeframe (purely technical time of 3 h versus 6.5 days for NGS) [18]. Moreover, the hands-on time can be drastically reduced to 5 min, and results can be obtained within one day, when FFPE sections or aspiration liquids are used directly [16], compared to other techniques (NGS, 4 h) [19]. A recent large-scale (221 patients) prospective bi-center study demonstrated a time-to-genotyping result of 1.9 days for the Idylla system versus 14.2 days for the NGS workflow (which encompasses DNA extraction) in real-life practice [20]. More importantly, TKI treatment was initiated in 7.7 +/− 1.2 working days for Idylla versus 20.3 +/− 6.7 for NGS [20]. Droplet digital PCR (ddPCR) is an alternative rapid method using small amounts of DNA with a higher sensitivity. This absolute quantification method partitions fragmented DNA molecules into individual oil droplets, allowing individual PCR reactions and droplet counting. Therefore, ddPCR droplet count closely reflects the composition of the nucleic acid mixture and allows the detection of infinitesimal quantities of mutated DNA. Hence, its sensitivity ranges from around 0.001% to 0.01%, while Idylla barely reaches 0.2 to 1% [21,22,23], even if these results, obtained in the situation of native DNA in sufficient quantity, are expected to differ with FFPE-extracted material producing high background noise. In addition, the turnaround time of PCR-based methods is less than one day longer than that of Idylla (real-life delay of 5.6 days versus 4.9 days in a large study on 780 colorectal cancer) [24].

Importantly, the establishment of sensitivity, linearity, limit of detection (LOD), or limit of blank (LoB) has mostly been carried out using control DNA of high quality and quantity, for both Idylla and ddPCR studies. In real-life situations, DNA is frequently extracted from small parts of FFPE biopsies, resulting in poor DNA quality and quantity. In addition, ddPCR and Idylla technologies have mainly been compared in the situation of hotspot mutation detection (such as KRAS or BRAF alterations), based on allelic discrimination (competition between hydrolyzable probes complementary to either the mutated or the unmutated sequence). However, a huge variety of exon 19 deletions/insertions exist (40 described in the TCGA, Appendix A), and drop-off probes have to be combined in the same reaction to cover the maximal number of alterations. This situation is expected to increase the risk of false negative (FN) results due to probe design and false positive (FP) results due to polymorphisms. In addition, no exhaustive study of del19 subtypes has been conducted yet.

We thus took advantage of a tumor sample collection comprising over 10,000 specimens to select 39 unique deletions of the exon 19 and fully address the question of sensitivity for del19 testing. Additionally, we challenged the specificity of both approaches by analyzing nine different SNPs in exon 19. We chose to compare the performance of the Idylla platform system with the ID-*EGFR*(b) SENSI-50-v3 kit (ID_Solution) for the ddPCR technique, as both methods assess the *EGFR* theranostic alterations (del19 and L858R). The present findings allow us to propose clear cutoff values for the clinical testing of actionable *EGFR* alterations (L858R and del19) in real-life samples. Finally, we took into account each mutation frequency among lung cancer patients to produce the adjusted theranostic sensitivity of each test.

## 2. Results

### 2.1. Sample Inclusion

Since L858R and L861 point mutations are detected by a single probe, testing one sample of each is sufficient to validate the probe design. However DNA quality/quantity and tumor content deeply impact sensitivity and specificity. We thus selected 24 samples with L858R representative of different qualities (tumor content, sample size, fixative history…), and one sample with L861Q. Due to the presence of several variants, the detection of del19 by drop-off approaches is much more challenging than point mutation detection. Hence, we identified 40 unique del19 among the 534 samples positive for del19 from 22 studies of The Cancer Genome Atlas Program (TCGA) on NSCLC (Appendix A). We interrogated our routine biological database, containing 625 samples bearing a del19, and identified 39 unique del19 with available material. Interestingly, 20 of these were not described in the TCGA database (corresponding to rare variants). These 39 deletions are representative of 95.1% of the del19 in the TCGA database alone, and of 93.4% of the joint cohort (local plus TCGA, N = 1159; Appendix A). Additional samples of various tumor cell content were included to reach 63 specimens with del19. Last, to fully challenge the specificity, we selected 10 samples with single nucleotide polymorphisms (SNP) in exon 19, in addition to 24 wild-type (WT) samples. In sum, 122 samples were included in this study, encompassing a highly comprehensive spectrum of del19 and SNP.

A major confounding factor in comparative studies is the biological heterogeneity of the samples, particularly in the case of FFPE samples. Different sections can be extracted at different time points, and DNA degradation increases with longer storage time in FFPE. Additionally, different extraction methods may be employed. Comparing FFPE ribbons to extracted DNA from other sections of the same sample is unsatisfactory. To overcome these biases, DNA was extracted from the same sections and partitioned between the two techniques at a fixed ratio of one per five, corresponding to 10 ng for ddPCR and 50 ng for Idylla, as commonly recommended by the manufacturer’s instructions and reference publication [25].

### 2.2. Limit of Blank (LoB), Positive Threshold, and Technical Validation Criteria

Considering ddPCR, we evaluated the background noise and quantified the FP droplets using 34 WT FFPE-extracted DNA samples. The highest percentage of mutated droplets (MDs) detected was 0.3% in both HEX (del19, L858R detection) and FAM (L861Q detection) channels (Figure 1). Applying the rule of three times the noise, we established the LoB at 1% of MDs. Based on our experience with ddPCR on FFPE samples, poor-quality DNA can produce up to three positive MDs even with a total positive droplet (TPD) count below 400, which would exceed the 1% cutoff. Therefore, we defined a sample as positive if it had at least five MDs and a variant allele frequency (VAF) above the LoB of 1%. Additionally, the minimal count of TPDs (which contains amplifiable material) recommended by the manufacturer for technical validation is 200 (approximately equivalent to 100 cells). We proposed to lower this threshold only for mutated samples. Among our samples identified as mutated by NGS, twelve (n°12, 22, 25, 28, 30, 31, 32, 47, 51, 57, 58, 63) demonstrated less than 200 TPDs (ranging between 62 and 194), with VAF ranging from 20.2% to 76.9% (with MDs > 4; see Appendix A). Consequently, the technical validation threshold was set at 200 TPDs for WT samples and 50 TPDs for mutated samples (Table 1). Notably, due to our technical criteria, the limit of cellularity (LOC) is de facto set at 2% (1% of VAF) for analyses with more than 400 TPDs, and it increases up to 16% for experiments with 50 TPDs (equivalent to 4 MDs for 50 TPDs). Thus, samples with infiltrated tumor cells and low material quantity should be interpreted with caution, as outlined below.

Idylla does not provide specific criteria for considering a sample as mutated, such as the detection of a Cq value for the mutated allele (Cq_mt_) or a threshold for the ΔCq (difference between the Cq values of the reference gene (Cq_ref_) and the Cq_mt_). In these conditions, it is not possible to analyze the limit of blank (LoB) or background noise. In our experience, the highest value of ΔCq in a mutated sample was 13.3, and the highest Cq_mut_ was 35. We tested and classified 62 known *EGFR*-mutated samples based on their detected Cq_ref_ values. Interestingly, the three samples (n°67, 51, and 63) with the highest Cq_ref_ values (30.1, 27.6, and 27.3, respectively) were identified as WT by the Idylla software (Version 4.3.0.380), despite being positive according to NGS analysis (Appendix A). The TPDs and the VAF determined by ddPCR were 4, 136, 84, and 75%, 61%, 20%, respectively. Surprisingly, sample 67 showed only four positive droplets in ddPCR due to a limited number of “amplifiable” DNA molecules, and it was not expected to reach a Cq value enabling technical validation. Sample 51, which had the correct genotype according to ddPCR with sufficient DNA quality and tumor cellularity, was likely not detectable by Idylla and was considered a technical FN. Sample 63 (Cq_ref_ = 27.3), in which the mutation was detected by Idylla in another sample with higher cellularity and DNA content, was classified as an FN due to low DNA quantity (FNq), similar to sample 67. As samples with Cq_ref_ values below 26 consistently demonstrated correct genotyping, we propose setting the maximum value of Cq_ref_ to 26, provided that the tumor content is sufficient.

### 2.3. Material Limitations and Method Validation

#### 2.3.1. DNA Quantity—Limit of Detection (LOD) and Limit of Quantification (LOQ)

The effect of decreasing DNA quantities (10 to 0.3 ng) of an L858R mutated sample (n°81) with low cellularity (estimated at approximately 20% based on histopathological observation) was assessed by ddPCR. A linear correlation was observed between the TPD count and the DNA quantity (Figure 2A), and the L858R mutation was detected in all conditions, with a VAF of around 10% (as expected) when at least 1.25 ng of DNA was present. However, when lower DNA quantities (0.63 ng and 0.31 ng) were loaded, the TPD count was below 200, and the precision of the VAF decreased (4.8% and 7.9% of VAF). Based on these results, we propose an LOD of 0.31 ng and an LOQ of 1.25 ng. In real-life samples, the recommended DNA quantity for ddPCR assay (ID_Solution) is 10 ng, and among the 10 samples with detectable del19 alteration and lower amounts of DNA (ranging from 3.15 to 9.6 ng), 9 were correctly genotyped by ddPCR (Figure 2B). The remaining sample, even though it exhibited three MDs (out of four TPDs), was classified as “not interpretable” according to our diagnostic thresholds (low TPD count).

For Idylla testing, only two samples with del19 (n°28 and 67) were analyzed using less than 50 ng of DNA (22.89 ng and 40 ng, respectively; Figure 2B). The first sample returned a positive result, while the second sample, with a Cq_ref_ value of 30.1, was classified as a FNq (as discussed above). To further investigate the limits of quantification, we loaded decreasing amounts of DNA (50 to 5 ng) from sample 81 (similarly to the ddPCR setup) and observed—as expected—a logarithmic correlation between the Cq_ref_ and the DNA quantity (Figure 2A). The mutation was still detected with 5 ng of DNA (Figure 2A), despite a Cq_ref_ slightly above 26. The LOD was thus set at 5 ng. The ΔCq parameter exhibited decreased precision below 25 ng, leading us to set the LOQ at 25 ng.

In conclusion, both methods are quantitative when using the dilution range of the same DNA sample. However, ddPCR is more than tenfold more sensitive, with an LOD of 0.31 ng compared to 5 ng for Idylla. In real-life samples, we recommend a minimum loading of 25 ng for Idylla and 1.25 ng for ddPCR, provided that there is sufficient cellularity (above 20%).

#### 2.3.2. Sample Tumor Content—Limit of Cellularity (LOC)

The estimation of tumor cellularity, evaluated by a pathologist, is an essential input parameter that determines the technical choice and, ultimately, the interpretation of the result. To avoid FN due to insufficient cellularity (FNc), it is necessary to determine the LOC. In order to do so, we selected 46 samples that were identified as mutated by both Idylla and ddPCR and technically validated (≥200 TPD or Cq_ref_ ≤ 26). These samples were classified based on histopathological estimated cellularity (<10%, 10–30%, 30–50%, and 50–80%). The ΔCq determined by Idylla and the VAF determined by ddPCR were analyzed (Figure 3A). As expected, the VAF increases with cellularity, which is in agreement with the absolute quantification capacity of ddPCR. Thus, the LOC corresponds to twice the LoB set at 1%.

However, no correlation was observed for the Idylla technology. Even when considering the VAF determined by ddPCR rather than the estimated tumor content as the gold standard (Appendix A). It is important to note that the reported Cq_mut_ values are not “real” but extrapolated from recalculated amplification curves, meaning that Cq values can be unrelated to material cellularity and mutation allelic frequency. In this scenario, a mathematical determination of the LOC cannot be made for Idylla testing. Therefore, we tested another approach and focused on the cellularity of the 13 samples returned as FN by Idylla (Figure 3B). For samples with histologically estimated tumor content below 30%, three returned as negatives (samples 74, 18, and 4) with ddPCR-estimated VAFs of 0.3%, 4.4%, and 1%, respectively. These results were considered as FNc, and we thus propose a cutoff of 5% (VAF), corresponding to the 10% cellularity recommended by Biocartis. The LOC was thus set at 10%. As a consequence, samples with higher cellularity (above 30%) and Cq_ref_ ≥ 26 were considered as technical FN (not detectable by the technology). Indeed, in these cases (sample numbers: 3, 8, 51, 56, 63, 67, 70, 71), the VAF estimated by ddPCR ranged from 20% to 61%, indicating a cellularity over 40%.

In conclusion, we established an LOC of 2% and 10% for ddPCR and Idylla, respectively, but without absolute confidence regarding Idylla. For routine use, we recommend a minimum percentage of tumor cells established by a pathologist at 10% for ddPCR and 30% for Idylla (mandatory in the case of no detected mutation).

#### 2.3.3. DNA Quality (Robustness)

Although we demonstrated the quantitativeness of both methods by using serial dilutions of the same DNA (Figure 3A), we failed to establish any correlation between the quantity of DNA loaded in each experiment and the Cq_ref_ obtained by Idylla or the TPD count obtained in ddPCR, when using real-life samples (Appendix A). The quality of DNA is an important parameter, particularly when working with FFPE samples. DNA fragmentation caused by oxidative damage induced by fixatives during storage hinders DNA amplification and affects quantification. To determine if the number of “amplifiable” DNA molecules is equivalent in both techniques, we plotted the TPD count observed after ddPCR analysis against the Cq_ref_ obtained with Idylla and found an excellent correlation (R^2^ = 0.86; Appendix A). This indicates that the robustness of both technologies can be considered equivalent.

### 2.4. Specificity

We analyzed 24 samples without variants in exon 19 and 21 of *EGFR* (Figure 4). As expected, no mutations were detected by either technique. To evaluate the risk of FP results, we examined 10 samples with 9 different SNPs located in exon 19 of the *EGFR* (G729R, E734K, P741S, V742I, L747S, L747P, R748T, A755P, and D761Y). The Idylla assay did not detect these SNPs, while ddPCR identified a signal for samples 41, 42, and 68 (corresponding to L747P, L747S, and V742I, respectively; Appendix A). Interestingly, the fluorescence signal for V742I was identified between the WT and L861Q channels, whereas the signals for L747P/S appeared between the WT and the del19 /L858R channels. Based on these distinct patterns, these SNPs could be identified and distinguished from the classical activating mutations del19/L858R or L861Q. In conclusion, both assays showed no FP results (N = 34) and achieved a specificity of 100%. However, the interpretation and selection of positive droplets after ddPCR with the ID_Solution kit require vigilance regarding the p.L747 and p.V742 positions.

### 2.5. Sensitivity and Adjusted Sensitivity—Spectra of del19 Detection

Hotspot mutations such as L858R and L861Q (N = 25) were identified with 100% sensitivity by both methods, provided that DNA quality/quantity and cellularity thresholds were respected. However, determining the sensitivity of del19 detection is touchier. We used 39 FFPE samples, each containing different deletions identified by NGS or Sanger sequencing (Figure 5). Sufficient material was available for both Idylla and ddPCR analyses in 31 samples, while 8 additional samples were exclusively analyzed by ddPCR (requiring only 1 to 10 ng).

Regarding Idylla, 21 out of 31 (67.7%) del19 samples were correctly genotyped. Among the 10 misdetected del19 samples (samples 3, 4, 7, 8, 17, 51, 56, 67, 70, 71), two had Cq_ref_ values > 26 (samples 51, 67) and were classified as FNq. One sample was classified as FNc (sample 4), and the remaining seven mutations (validated by NGS), were not listed in the TCGA database and were considered as technical FN, as they were not predicted to be covered by the Idylla PCR platform (c.2217_2234dup, c.2232_2248delinsTAAAATTC, c.2235_2241delinsAATTCCC, c.2236_2248delinsCTTC, c.2239_2260delinsCAAC, c.2248_2276delinsCCGAA, c.2252_2276delinsA; Appendix A). Interestingly, among the 25 undescribed del19 included in this study, Idylla was able to detect four unknown mutations despite no specific probes (sample 14—c.2235_2253delinsAACC; sample 16—c.2236_2248delinsCAAC; sample 27—c.2237_2251delinsTTG; sample 47—c.2239_2250delinsCCG).

In the case of ddPCR, 92.3% (36 out of 39) of the del19 samples were detected. Considering the three misdetected samples (3, 70, 71), all had sufficient TPDs and cellularity over 30% and were considered as technical FN. These samples were also misdetected by Idylla. Sample 70 harbored a novel deletion/insertion, while samples 3 and 71 exhibited known del19 deletions. When considering the common samples analyzed by both techniques, the sensitivity of ddPCR was 90.3% (28 out of 31), compared to 67.7% for Idylla. Among the identified deletions, 19 were not predicted to be detectable by the ID_Solution kit (Appendix A), while the 20 predicted deletions were accurately retrieved. As VAF is a continuous variable available for negative and positive samples (whereas ΔCq was not available for negative sample), the ROC curve was calculated with an AUC of 0.97 (Appendix A).

Given the significantly different prevalence of each del19 in lung cancer, it was crucial to consider the adjusted sensitivity (Ad-Se) within the entire lung cancer population. Firstly, we calculated the incidence of each deletion in the combined dataset (local + TCGA; Appendix A). The 39 (ddPCR) and 31 (Idylla) unique del19 analyzed represented 93.4% and 91.5%, respectively, of all the del19 in the combined database. By cumulating the frequencies of the detected deletions, we reported an Ad-Se of 92.58% for ddPCR and 89.91% for Idylla for the detection of del19. Now, regarding the overall theranostic sensitivity, the incidences of L858R/L861Q alterations and del19 in our cohort were found to be 7% and 6.2%, respectively. Considering a sensitivity of 100% for L858R/L861Q and 92.58% and 89.91% for del19, the adjusted theranostic values are 96.51% and 95.26% for ddPCR and Idylla, respectively.

In summary, ddPCR demonstrated a broader range of del19 detection compared to the Idylla assay (92.3% versus 67.7% sensitivity for unique del19). However, when considering the entire NSCLC population, both the Ad-Se and, more importantly, the theranostic sensitivity were equivalent.

## 3. Discussion

Precision medicine has become a standard practice, particularly in the case of lung cancer. Due to the rapid progression of these tumors [14], the current challenge is to accelerate molecular testing for patients with metastatic disease to rapidly initiate the appropriate therapy. In this study, we compared the performance of the two most commonly used rapid testing strategies: ddPCR and Idylla. The technical performances of ddPCR clearly surpass those of Idylla: reliable quantification, lower DNA input requirement (1 ng versus 10 ng), lower LOC (2% versus 10%), lower LOD (0.31 ng compared to at least 10 ng), and a much wider range of del19 detection (92.3% versus 67.7%). However, the Ad-Se and theranostic values are equivalent to those of the Idylla platform (assuming sufficient cellularity and DNA quantity). Idylla also benefits from the CE-IVD label and is implementable in pathology departments by technicians without molecular biology training, requires less technical handling, and is one day faster than ddPCR when using FFPE ribbons or sections instead of extracted DNA. Nevertheless, Idylla is a low-throughput, non-quantitative, and expensive technology [26], using much more biological material compared to ddPCR.

One of our objectives was to propose a reliable routine diagnostic algorithm based on fixed thresholds and to highlight the technical limitations of these techniques. We chose to compare the same material extracted from the same FFPE sections to ensure conclusive comparisons. The first parameter to ensure is the analysis of a sufficient number of DNA molecules to account for tumor heterogeneity and stromal cell contamination, especially in cases with low tumor content. In the case of Idylla, we propose a Cq_ref_ threshold of 26 to avoid the risk of FNq. Khalifa et al. reported 13 FN in a cohort of 65 pre-extracted DNA samples from NSCLC, all with Cq_ref_ above 23. In particular, low DNA input (5 ng) resulted in FN with Cq_ref_ above 26 [25]. The authors recommend reducing the threshold to 23, as do Grant et al. [27], who also used pre-extracted DNA from FFPE. We suggest cautiously validating WT results between Cq_ref_ 23 and 26 and considering results with Cq above 26 as non-interpretable. For ddPCR, a threshold of 200 TPDs is recommended, corresponding to 100 cells analyzed (assuming diploid cells). Based on our results, we propose lowering this threshold to 50 in the case of mutated samples. Considering a LoB of 0.9% and a background noise of up to four MDs, we set the following combination: VAF > 1% and MD > 4 as the positive cutoff. Other publications have reported similar LoBs of 0.4% [28] and absolute counts of FP droplets ranging from 3 to 5 [29,30,31].

Regarding DNA quantity, based on the dilution range of a real-life sample with low cellularity, we set the LOD at 5 ng and the LOQ at 25 ng for Idylla testing. De Luca et al. achieved excellent sensitivity using 10 ng of pre-extracted DNA (N = 32), although they manually recalculated the ΔCq value provided by the Idylla platform [32]. Grant et al. demonstrated that analytical sensitivity remains stable from 100% to 98.46% but drops to 90.77% when DNA input decreases from 250 ng to 50 ng and 20 ng [27]. Khalifa et al. also reported a sensitivity drop from 88.6% to 47.4% when using more or less than 50 ng [25]. Based on these findings, we recommend using 50 ng as input and not interpreting results with less than 10 ng (assuming tumor content is above 30%). Moving on to ddPCR, we report an LOD of 0.32 ng and an LOQ of 1 ng. Previous studies have established an LOD of 0.1 ng and an LOQ of 0.33 ng using DNA of good quality (cell line extract) [33]. When using FFPE-extracted controls, the use of 15 ng is recommended to detect mutations at a VAF of 1% [29]. Overall, we recommend an input range of 1 to 10 ng for ddPCR, but without any strict restrictions as long as technical cutoffs (TPD and VAF) are validated.

Additionally, the more DNA molecules are analyzed, the lower the tumor content can be. Therefore, thresholds should be set in relation to the LOC. While we demonstrated a clear correlation between histologically determined cellularity and VAF determined by ddPCR, we failed to establish any correlation with the ΔCq determined by Idylla, which is supposed to reflect tumor content. The LOC was *de facto* set at 2% for ddPCR as it aligned with the LoB of 1% (three times the background noise of 0.3%). This cutoff was confirmed using real-life samples with low cellularity. In line with this, Williamson et al. investigated VAF from 10% to 0.01% using 250 to 15.6 ng of DNA for each tested VAF and found that the LOC increases from 0.01% to 1% when using 250 ng or 15.6 ng as input [29]. This confirms our cutoff of 1% for loading 10 ng of DNA.

Furthermore, we determined an LOC for Idylla by analyzing the tumor content in FN results. Based on this analysis, we established a cutoff of at least 10% tumor content, as recommended by Idylla and Bocciarelli et al. [34]. Other publications focused on FFPE sections have reported lower detection thresholds, ranging between 1% and 5% VAF [35,36,37], but with Cq_ref_ values ranging from 20 to 23 [36]. Notably, larger multicenter studies have set input criteria of 10% [38], 20% [25], and even 30% [39] tumor content for Idylla. This variation may reflect differences in tumor content determination among pathologists, which becomes particularly relevant in cases of low cellularity and when studies are conducted prospectively across multiple centers. For routine use, we recommend input criteria of 30% tumor content for Idylla and 10% tumor content for ddPCR. Results should not be validated below 10% tumor content for Idylla and 2% tumor content for ddPCR, and WT results between these values should be interpreted cautiously. In a large-scale prospective study, the inclusion criteria based on cellularity (20%) and quantity (50 ng) resulted in the immediate invalidation of 16.3% of samples, with 11 out of 29 cytology samples being ineligible [25]. Other studies on cytological specimens reported positive detection but with high DNA content (Cq_ref_ > 23) and highlighted the importance of manual analysis of Cq curves [36,40]. Therefore, it is expected that a significant portion of tests conducted on samples with limited tumor material (such as biopsies or cytological specimens) will yield non-informative results with Idylla. More specifically, negative results should be interpreted with caution in this context.

In addition to cytologically determined tumor content, another crucial input criterion that can vary is DNA quantification. While we observed a linear correlation between DNA input and Cq_ref_ or TPD on a dilution range from the same FFPE-extracted DNA, indicating the quantitative potential of both technologies, the DNA quantity did not correlate with the actual number of amplifiable DNA molecules (TPD and Cq_ref_). This is likely due to variations in fresh sample preparation, fixation, and subsequent DNA fragmentation. Khalifa et al. observed a tendency for an inverse correlation (R^2^ not provided) between DNA concentration and Cq_ref_ using the Idylla system, possibly because they had a much larger range of concentrations (N = 577) [25]. Most interestingly, we report a clear correlation (R^2^ = 0.86) between Cq_ref_ and TPD, demonstrating that both parameters reflect the actual content of “amplifiable” DNA molecules.

Regarding the lack of informativeness of the ∆Cq parameter, we also failed to observe any correlation between the VAF determined by ddPCR and ∆Cq, similar to what has been reported for the *EGFR* T790M mutation [41]. However, another laboratory investigating KRAS testing from ctDNA analysis reported a correlation [42]. The quality of DNA used as input may be a factor in this discrepancy, as FFPE-extracted DNA is of poor quality, and the recalculation method used by the Idylla system may amplify variations due to degraded DNA. Precise quantification of VAF is not mandatory in the context of *EGFR* mutant lung adenocarcinoma, as these mutations are commonly driver mutations and not subclonal. However, it may be important for other applications, such as monitoring patient response to therapy using ctDNA [42].

Now, considering specificity, most studies conducted on FFPE or pre-extracted DNA in NSCLC, have reported no FP results for Idylla. Most interestingly, to our knowledge, this is the first study to investigate the impact of SNPs on exon 19. We believe this is a significant consideration for diagnosis, especially when using drop-off probes, as SNP presence can destabilize hybridization and potentially lead to FP results. We investigated nine unique SNPs in exon 19 and demonstrated no FP for Idylla, while ddPCR detected two positions (p.L747 and p.V742). Although the generated droplets can be differentiated from the pathogenic ones, caution is necessary to avoid misinterpretation. If careful droplet selection is ensured, both techniques achieve 100% specificity.

No L858R mutations were misdetected, with both testing methods achieving 100% sensitivity for this point mutation detection. Most studies on point mutation detection in NSCLC primarily focus on the resistance mutation T790M. Lee et al. reported 5 out of 19 FN (with cellularity below 5%) and a sensitivity of 74% for Idylla, while ddPCR achieved 100% sensitivity [41]. Other publications have also reported a similar poor sensitivity for T790M detection, with five and nine missed patients in different studies [15,43]. Furthermore, when screening 39 unique del19 mutations, we found herein 10 FN, 7 of them attributed to probe design issues, resulting in a technical sensitivity of 67.7% for Idylla. In a prospective bi-center study, 13 out of 54 (non-unique) del19 mutations were misdetected, resulting in an overall analytical sensitivity of 75.9%, which is consistent with the present findings [25]. In real-life scenarios, after normalizing for the incidence of each del19, we report herein an Ad-Se of 89.91%. The discrepancy between theoretical and real-life values is due to FNq, FNc, and LOQ, without considering design failures. In comparison, ddPCR testing design performs much better, with 92.3 (36/39) unique del19 mutations detected, resulting in a 92.58% Ad-Se. We also identified 4 and 15 new mutations not predicted to be detected by Idylla and ddPCR, respectively (Appendix A). However, considering the frequency of theranostic mutations in NSCLC, the adjusted theranostic values of the assays are very similar, with 96.51% and 95.26% for ddPCR and Idylla testing, respectively.

Lastly, in terms of cost, a previous study has identified a cutoff of 110 samples per year favoring ddPCR over Idylla and other rapid techniques such as BEAMing and COBAS z48026 [26]. This corresponds to two samples per week, indicating that Idylla is suitable for urgent situations, but not for routine *EGFR* testing. In our assessment, we found that the cost of Idylla was approximately twice that of ddPCR when starting from extracted DNA (see Table 1 and material and method section). On the other hand, ddPCR can accommodate low quantities of DNA and even lower cellularity compared to NGS, making it well-suited for routine conditions for biopsy or cytology samples. Still, both techniques should be followed by a more comprehensive NGS analysis to detect rare or novel mutation variants associated with the risk of FN, as well as other targetable alterations in different genes. Additionally, the mechanisms of resistance observed in patients are complex and heterogeneous, necessitating large-scale genomic investigations [44].

## 4. Materials and Methods

### 4.1. Samples

Human biological samples were obtained from the *Tissus-Tumorothèque Est* Biobank (CRB-HCL *Hospices Civils de Lyon* BB-0033-00046), authorized by the French Ministry of Research. The protocol was approved by the Ethics Committee of the HCL (June 2019, identification number: 2018-A00680-55) and the French data protection commission (CNIL, authorization 919361). Molecular results are issued from routine clinical testing of 10,380 NSCLC tissues between 2011 and 2018. We included 63 samples among the 625 with del19, corresponding to 39 unique deletions; 24 samples with L858R, and one sample with L861Q. Additionally, 34 wild-type samples were included as controls. From 2009 to March 2016, mutation status was determined by Sanger sequencing of exons 18-19-20 and 21, and after March 2016 by next-generation sequencing (NGS) with a custom amplicon panel (BED of 22 kb, covering EGFR exons 2-7-9-18-19-20 and 21) on Ion Personal Genome Machine (PGM Ion Torrent, ThermoFisher Scientific, Waltham, MA, USA). The bioinformatic analysis was carried out on ionreporter. The diagnoses of NSCLC were established by a pathologist according to the World Health Organization Classification of 2015. Histological tissue samples were fixed in 10% neutral buffered formalin (4% formaldehyde) following standard guidelines for immunohistochemical testing. The sample cellularity was blindly determined on hematoxylin and eosin-stained sections by two expert pathologists before DNA extraction.

### 4.2. DNA Extraction and Mutation Analysis

Serial sections were used for DNA extraction. The Maxwell^®^ RSC RNA FFPE Kit (AS1440, Promega, Charbonnières-les-Bains, France) was used (without DNAse) to obtain total nucleic acid (TNA). The FFPE samples were deparaffinized and scraped or microdissected with a laser (Leica DM6000B, Leica, Solms, Germany) to specifically select areas enriched in tumor cells (2 tubes containing about 500,000 μm^2^ of tumor cells each). Considering microdissection, tumor content is expected to be superior to 50%. DNA extraction tubes with scrapped or microdissected samples were prepared following the manufacturer’s instructions and were incubated with Proteinase K overnight at 56 °C. Subsequently, the extracted DNA was quantified using the QuantiFluor(R) ONE dsDNA System (E4871; Promega) according to the manufacturer’s instructions. A quantity of 10 nanograms and 50 nanograms were used for ddPCR and Idylla assays, respectively. In cases of insufficient material, the DNA was split while maintaining the same proportions (1 part to 5 for ddPCR and Idylla, respectively) to ensure comparable assay conditions.

To run Idylla assays, 50 ng of extracted DNA was directly loaded into the Idylla *EGFR* Mutation test (A0060) cartridge. Quantitative PCRs were run on the Idylla platform, output data were provided by the Idylla embedded software (version 4.3.0.380); the quantification cycle value of a reference gene Cq_ref_ and when mutated, the ΔCq value (Cq of the mutated amplicon Cq_mut_ gene minus Cq_ref_), the technical validation, and the genotype. No manual analysis was undertaken.

The detection of *EGFR* hotspot mutations (del19, L858R, L861Q) by digital PCR was performed using the multiplex digital PCR kit *EGFR*-50Sensi-V3 (ID-Solutions), following the manufacturer’s instructions. Probes with HEX reporter fluorophore were used to amplify del19 and L858R, and while the L861Q mutation could be detected in the FAM channel, the intermediate channel was dedicated to wild-type allele. Selection of droplets for genotype assessment was undertaken by overlaying all the results of one experiment (>10 samples), containing at least one positive control, so that selection can be considered as blinded (even if the genotype was known). Digital droplets were generated using the QX200 AutoDG Droplet Digital PCR System (Biorad). PCR amplifications were run in a standard thermal cycler, and the fluorescent positive and negative droplets were counted using the QX200 Droplet Reader and analyzed with the Quantasoft Analysis Pro software (version v1.0.596, Biorad, Marnes-la-Coquette, France). All the experiments were carried out on the Biogenet Est platform LBMMS CBE (HCL).

### 4.3. In Silico Analysis

Data from the TCGA database were extracted through the cBioportal portal. Patients with NSCLC or LUAD were selected from the 22 available studies (luad_broad, luad_cptac_2020, luad_msk_npjpo_2021, luad_mskcc_2015, luad_mskcc_2020, luad_mskimpact_2021, luad_oncosg_2020, luad_tcga, luad_tcga_pan_can_atlas_2018, luad_tcga_pub, luad_tsp, lung_msk_2017, lusc_cptac_2021, lusc_tcga, lusc_tcga_pan_can_atlas_2018, lusc_tcga_pub, nsclc_mskcc_2015, nsclc_mskcc_2018, nsclc_pd1_msk_2018, nsclc_tcga_broad_2016, nsclc_tracerx_2017, nsclc_unito_2016). Among these data, samples with in-frame del19 mutations were selected, and duplicate samples were removed to obtain the total panel of described del19 mutations in the TCGA database (N = 534, encompassing 40 unique del19). Finally, the frequency of each deletion *f_i_* was calculated as follows: *f_i_* = n_i_/N, where n_i_ is the number of samples bearing the unique (i) del19 among the database, and N is the total number of samples bearing a del19 in the database. Calculations were made for the TCGA dataset (N = 534) and for the common dataset encompassing our local database (N = 1159). The adjusted sensitivity (Ad-Se) was calculated by adding the frequency (∑*f_i_*) of each unique del19 detected among the compiled dataset (or among the TCGA cohort alone). The global theranostic sensitivity was calculated as follows: Th.Se. = Ad-Se_del19_ × Prev_del19_ + Se_L858R_ × Prev_L858R_ + Se_L861Q_ × Prev_L861Q_)/(Prev_del19_ + Prev_L858R_ + Prev_L861Q_), where Prev stands for prevalence in the lung cancer population.

### 4.4. Graphs and Statistics

Graphs and charts were carried out using Excel 2016 Microsoft Office™ and R 4.1.0 (R Core Team, 2021), R: A language and environment for statistical computing, the ROC curve was obtained using the ROCR package. The sensitivity was calculated as follows: Se = P/TP, representing the number of samples detected as mutated (P) among the true positive samples (determined by NGS/Sanger). The specificity was calculated as follows: Spe = N/TN, meaning the number of samples detected as unmutated among the true negative samples (determined by NGS/Sanger).

### 4.5. Cost and Hands-On Time

A cost analysis was conducted for both single-patient and multi-patient scenarios, considering the expenses associated with hands-on time and reagent kit costs. Notably, this calculation excludes the costs of consumable materials not included in the kit, equipment procurement, and maintenance. The hourly rate for technical labor was set at EUR 41/h, while biological interpretation was valued at EUR 97/h, following French regulations. For the Idylla system, the estimated technical time required was 10 min, encompassing DNA deposition and cartridge preparation, as well as the initiation of the analysis on the Idylla instrument. Biological interpretation took an additional 15 min for visualization and curve validation. The cartridge cost EUR 220. For ddPCR, the total cost was determined based on 1.4 h of hands-on technical work, which included PCR mix preparation and droplet generation (45 min), droplet reading (20 min), and droplet selection for genotyping using Quantasoft Analysis Pro software (20 min). Additionally, 20 min of pathologist time was considered, and reagent costs ranged from EUR 60 to EUR 24 for 1 to 10 simultaneous analyses (regardless of the number of samples, both negative and positive controls were required for each run). Importantly, the ddPCR experiment allowed for the management of several samples within 5 h of hands-on time, whereas the Idylla device could process only one sample in 3 h.

## 5. Conclusions

In conclusion, both technologies are valuable in urgent or pre-screening scenarios, serving as a preliminary step before undergoing a full NGS screening. We propose herein a diagnostic algorithm to ensure secure testing. Provided that the proposed thresholds are applied, both technologies offer comparable theranostic value. Idylla technology should be restricted to pathology departments without access to molecular platform facilities, while ddPCR, which can accommodate low quantities and cellularity, presents a more reliable and flexible alternative for rapid testing.

## Figures and Tables

**Figure 1 ijms-24-15684-f001:**
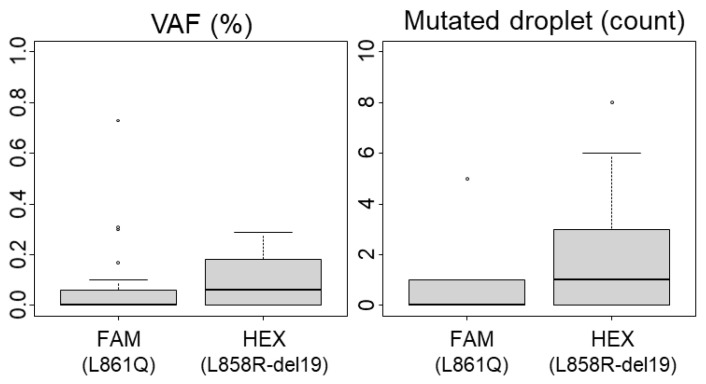
**Limit of Blank (LoB) for ddPCR analysis**. The variant allelic fraction (VAF) and the number of mutated droplets were determined using 34 wild-type (including polymorphisms) samples after performing ddPCR analyses. Boxplots representing mediane and quartiles (75%, 25%) are show, outliers are depicted by circles The L861Q mutation is detected in the FAM channel and L858R mutations or exon 19 deletions are detected in the HEX channel.

**Figure 2 ijms-24-15684-f002:**
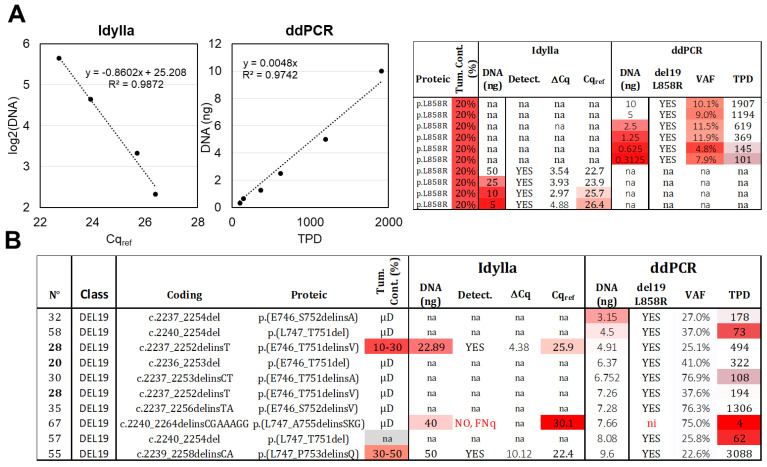
**Limit of detection (LOD) and limit of quantification (LOQ).** (**A**) Serial DNA dilutions were made from the same DNA extract of an FFPE sample with 20% tumor content, bearing an L858R alteration. These dilutions were analyzed using Idylla (ranging from 50 ng to 5 ng) and ddPCR (ranging from 10 ng to 0.31 ng). Correlations were established between the DNA quantity and Cqref or TPD, two parameters related to amplifiable DNA molecules. The raw data are shown on the right (VAF: variant allelic fraction in percentage, TPD: total positive droplet count, Tum. Cont.: tumor content in percentage, na: not available). (**B**) Analysis of tumoral samples with low DNA input and del19 alteration. Thirteen samples below 10 ng were analyzed using ddPCR, while two samples below 50 ng were analyzed using Idylla. Gradient of red in box represents samples with values exceeding the established thresholds. Samples extracted *after laser microdissection* (µD) have a content of tumoral cells > 50%. ni: not interpretable, FNq: false negative due to low DNA quantity. Samples from the same patient are indicated in bold.

**Figure 3 ijms-24-15684-f003:**
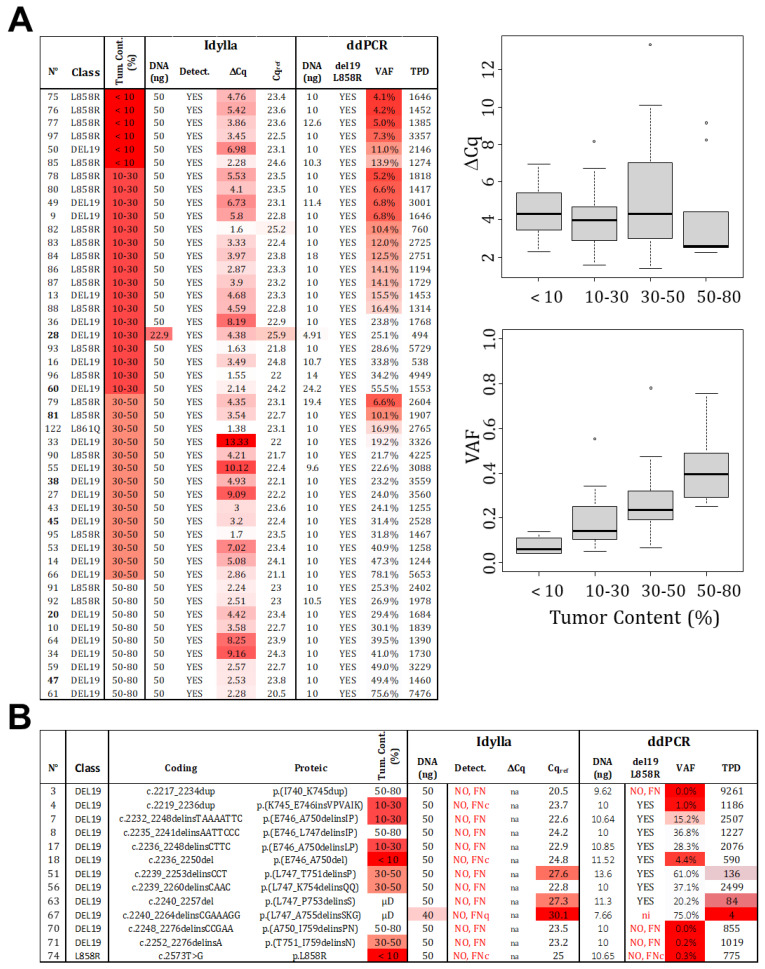
**Limit of cellularity (tumor content).** (**A**) A total of 46 samples, identified as mutated (L858R or del19) by both technologies, were categorized based on the estimated tumor content (detailed data on the left). While a positive correlation is observed between VAF and tumor content determined by ddPCR, no such correlation is visible for ∆Cq and tumor content detected by Idylla (right panels). Distribution of samples is represented by point and the boxplot (median and standard deviation). (**B**) False negative results were analyzed and classified as FNc (due to low cellularity), FNq (due to low quantity), or FN due to technical failure to detect the mutations (FN). (Gradient of red box represents samples with value exceeding our thresholds, VAF: variant allelic fraction in percentage, TPD: total positive droplet count, Tum. Cont.: tumor content in percentage, µD: samples which underwent laser microdissection to specifically select tumor cells).

**Figure 4 ijms-24-15684-f004:**
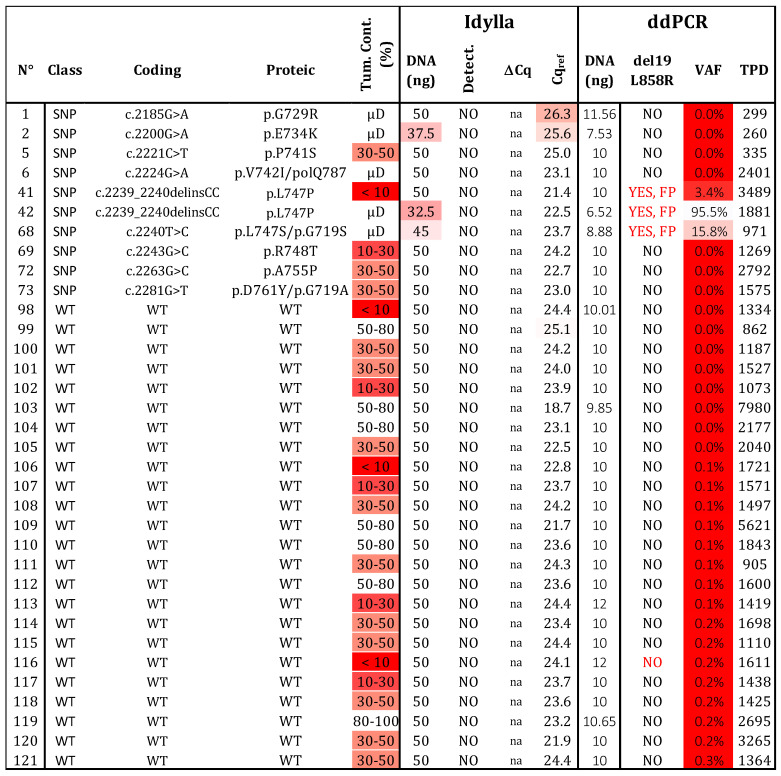
**Specificity determination.** Thirty-four samples, determined to be wild type (N = 24) or bearing a polymorphism (SNP; N = 10) by NGS, were assessed using both testing strategies. Three variants at positions p.747 and p.742 produced a signal in ddPCR, but they can be distinguished from real mutated signals (see Appendix A). WT: wild type, SNP: Single nucleotide polymorphism, VAF: variant allelic fraction in percentage, TPD: total positive droplet count, Tum. Cont.: tumor content in percentage, µD: microdissected sample, FP: false positive. Gradient of red cases represents samples with data exceeding the established thresholds for Idylla (Cqref > 26 and DNA < 50 ng) and ddPCR (TPDs < 200, DNA < 5 ng, or VAF < 50%.

**Figure 5 ijms-24-15684-f005:**
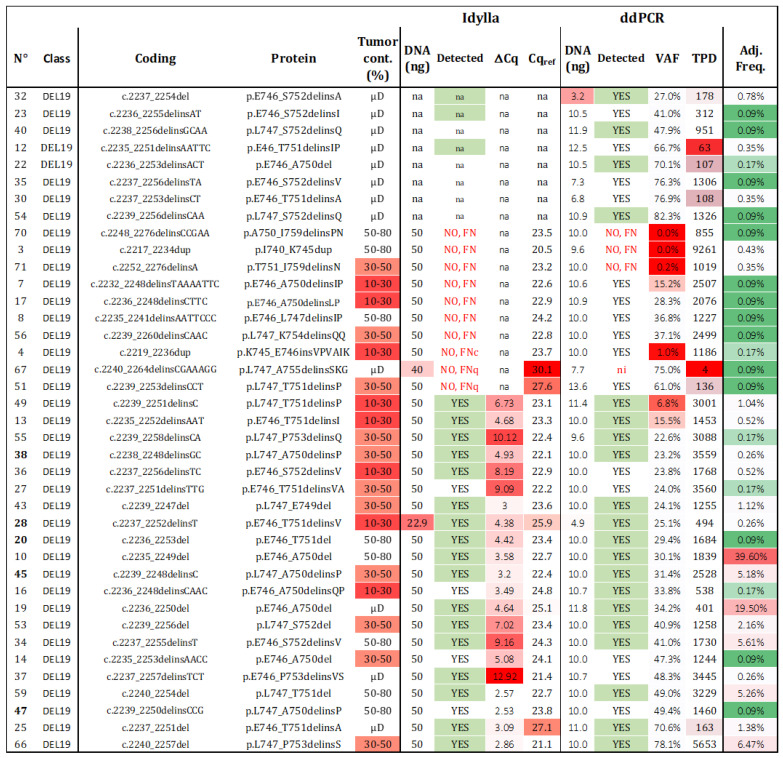
**Sensitivity of del19 detection.** Thirty-nine samples with a unique del19 alteration were assessed using both techniques. The detection results “YES”, “NO”, “FN” (false negative), “na” (not available), and ni (not informative) are summarized for each tested alteration, with those predicted to be detected highlighted in green. The frequency of each alteration among all del19 detected in patients from our cohort and the TCGA was calculated (VAF: variant allelic fraction in percentage, TPD: total positive droplets, Tum. Cont.: tumor content in percentage, µD: microdissected samples, Adj. Freq: adjusted frequency). Red cases represent samples with data exceeding the established thresholds for Idylla (Cqref > 26 and DNA < 50 ng) and ddPCR (TPD < 200, DNA < 5 ng, or VAF < 50%). Adjusted frequencies are highlighted following a color gradient (low frequencies in green and high frequencies in red).

**Table 1 ijms-24-15684-t001:** Costs, Technical and biological cutoffs for clinical testing of FFPE samples.

	*ddPCR*		*Idylla*	
*Technical validation threshold*	≥200 TPDs *		Cq_ref_ ≤ 26	
*Limit of blank (LoB)*	1% VAF ** and ≥5 MDs		*na*	
*Limit of detection (LOD)*	0.31 ng		5 ng	
*Limit of quantification (LOQ)*	1.25 ng		25 ng	
*Limit of cellularity (LOC)*	2% (twice the LoB)		10% (ΔCq unrelated to cellularity)	
*Recommended DNA input*	10 ng (up to 1 ng for high TC)		50 ng (up to 10 ng for high TC)	
*Recommended TC input*	10%		30%	
	L858R	del19	L858R	del19
*Specificity (Spe)*	100%	100% ***	100%	100%
*Sensitivity (Se)*	100%	92.3%	100%	67.7%
*Adjusted sensitivity*	100%	92.58%	100%	89.91%
*Theranostic sensitivity*	**96.51%**		**95.26%**	
*Total cost for 1 to 10 analysis*	€251		€150–€114	

TC: tumor content; TPDs: total positive droplets; MDs: mutated droplets; VAF: variant allele frequency; * Up to 50 in case of mutated sample; ** up to 8% in case of 50 TPDs; *** caution for p.747 and p.742 positions; na: not available.

## Data Availability

Data is contained within the Appendix A.

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
