# Peer review of "Comparison and Validation of Rapid Molecular Testing Methods for Theranostic Epidermal Growth Factor Receptor Alterations in Lung Cancer: Idylla versus Digital Droplet PCR"

_ijms, 2023, doi:10.3390/ijms242115684_

Round 1

Reviewer 1 Report

The study is relevant, well-designed and meticulously performed. I have only a few comments.

The conclusions in the abstract are slightly different than those in the text: at least to me, they sound more strongly pro-ddPCR than the rest of the text. I would edit to make them homogeneous, perhaps mirroring the Conclusions at the end of the manuscript.

L79. I would mention that Idylla can also be used directly on cytological material with a similarly short hands-on time (D'Ardia A, et al. Advanced non-small cell lung cancer: Rapid evaluation of EGFR status on fine-needle cytology samples using Idylla. Pathol Res Pract. 2021 Aug;224:153547. doi: 10.1016/j.prp.2021.153547 )

L83 "NGS20" is this a misformatted citation?

L81 I would also mention that same-day results can be obtained (D'Ardia et al (see above), as well as De Luca C, et al. Rapid On-site Molecular Evaluation in thyroid cytopathology: A same-day cytological and molecular diagnosis. Diagn Cytopathol. 2020 Apr;48(4):300-307. doi: 10.1002/dc.24378. )

L89: the difference in sensitivity of ddPCR (vs Idylla) is striking, but are these numbers clinically relevant? Does this have any impact, given the higher LoB?

L90: I am confused. How can the turnaround time of ddPCR be less than one day longer than idylla (5.6 vs 4.9 days) when a few lines before the authors state that the TAT of idylla is 1.9 days, and some experiences (see above) mention same-day results?

Author Response

The study is relevant, well-designed and meticulously performed. I have only a few comments.

We would like to express our thanks to the reviewer for their valuable advice and meticulous proofreading

The conclusions in the abstract are slightly different than those in the text: at least to me, they sound more strongly pro-ddPCR than the rest of the text. I would edit to make them homogeneous, perhaps mirroring the Conclusions at the end of the manuscript.

Indeed, the abstract may appear somewhat partial; therefore, we attempted to align it more closely with the conclusion by replacing..

L 35 : “ddPCR performs much better for small specimens and low tumoral content, and should be preferred in this context.”

by

“ddPCR performs better for small specimens and low tumoral content, in other situations Idylla is an alternative (especially if molecular platform is absent).”

L79. I would mention that Idylla can also be used directly on cytological material with a similarly short hands-on time (D'Ardia A, et al. Advanced non-small cell lung cancer: Rapid evaluation of EGFR status on fine-needle cytology samples using Idylla. Pathol Res Pract. 2021 Aug;224:153547. doi: 10.1016/j.prp.2021.153547 )

It is effectively an important point, we have modified the text as bellow

L 75 :” The Idylla cartridge, composed of microfluidics and preloaded reagents, accepts either formalin-fixed-paraffin-embedded tissue (FFPE) ribbon or extracted DNA, and carries out deparaffinization, DNA extraction, and PCR amplification [15,17] »

to

L75 : “The Idylla cartridge, composed of microfluidics and preloaded reagents, accepts either formalin-fixed-paraffin-embedded tissue (FFPE) ribbon, extracted DNA or biological fluids (such as fine-needle aspiration liquid[16]), and carries out deparaffinization, DNA extraction, and PCR amplification [15,17]. »

L83 "NGS20" is this a misformatted citation?

> We thank the reviewer for his accurate proofreading, the citation has been correctly formatted (L86)

L81 I would also mention that same-day results can be obtained (D'Ardia et al (see above), as well as De Luca C, et al. Rapid On-site Molecular Evaluation in thyroid cytopathology: A same-day cytological and molecular diagnosis. Diagn Cytopathol. 2020 Apr;48(4):300-307. doi: 10.1002/dc.24378. )

> we agree, this is now mentioned at lane 78 : “to detect EGFR mutations in a quick timeframe (3 hours versus 6.5 days for NGS) “

> We emphasized the importance of cytology by revising the following sentence as shown below:

L79 : “Moreover, the hands-on time can be drastically reduced to 5 minutes, and results can be obtained within one day, when FFPE sections, or aspiration liquids are used directly[16], compared to other techniques (NGS, 4 hours) [19]. »

L89: the difference in sensitivity of ddPCR (vs Idylla) is striking, but are these numbers clinically relevant? Does this have any impact, given the higher LoB?

We concur that this point was inadequately introduced. Consequently, we have revised the sentence as follows:

L89 : “Therefore, ddPCR droplet count closely reflects the composition of the nucleic acid mixture and allows the detection of infinitesimal quantities of mutated DNA, its sensitivity ranging from around 0.001% to 0.01%, while Idylla barely reaches 0.2 to 1%[21–23].

To

L89 :“Therefore, ddPCR droplet count closely reflects the composition of the nucleic acid mixture and allows the detection of infinitesimal quantities of mutated DNA. Hence its sensitivity ranges from around 0.001% to 0.01%, while Idylla barely reaches 0.2 to 1%[21–23], even if these results, obtained in situation of native DNA in sufficient quantity, are expected to differ with FFPE-extracted material producing high background noise.”

L90: I am confused. How can the turnaround time of ddPCR be less than one day longer than idylla (5.6 vs 4.9 days) when a few lines before the authors state that the TAT of idylla is 1.9 days, and some experiences (see above) mention same-day results?

> Yes, we agree it can be confusing :

- the strictly minimal “technical time” for Idylla is about 3h long (if no DNA extraction is done), and for ddPCR of 5h. If DNA need to be extracted, one day is to be added, so about 2 days for ddPCR and 1.5 day for Idylla.

We added a section in the material and methods (L 608), to explain this point :

L625 : “Importantly, the ddPCR experiment allowed for the management of several samples within 5 hours of hands-on time, whereas the Idylla device could process only one sample in 3 hours”

> Various reports have assessed the 'average real-life time' (different from technical time) through retrospective analysis of result publication delays in pathology departments. Laboratories may incorporate an extraction step and/or allocate a specific day per week for this technology. As a result, the overall delay can extend beyond one or two days, and this duration may vary between laboratories. However, the time difference between the two technologies remains approximately half a day.

L78 : “to detect EGFR mutations in a quick timeframe (3 hours versus 6.5 days for NGS)” è “(purely technical time of 3 hours versus 6.5 days for NGS)”

L95 : “…of Idylla (5.6 days versus 4.9 days in a large study on 780 colorectal cancer” è “of Idylla (real-life delay of 5.6 days versus 4.9 days in a large study on 780 colorectal cancer..”

Reviewer 2 Report

-Considering that two methods, Idylla and ddPCR, have been compared in this study, Idylla should be used instead of rapid molecular tests methods instead of ddPCR in title.

-Like de19, which shows the position of gene deletion in exon 19. State the location of the l858r mutation in exon 21.   -In the abstract, the sum total of the mentioned samples is 121, while the set of purified DNAs is 122.   -How were the 34 wild type negative samples confirmed not to carry any of the mutants?   -A table should be added and positive samples should be shown based on the type and number of their mutations.   -Statistical calculations in the calculation of sensitivity and specificity in the diagnosis of patients should be well described. The ROC curve should be  displayed.  

-The comparison of the two Idylla and ddPCR methods in terms of the time required to perform the test and its cost should be added in the article.

-

Author Response

-Considering that two methods, Idylla and ddPCR, have been compared in this study, Idylla should be used instead of rapid molecular tests methods instead of ddPCR in title.

> We thank the reviewer for this suggestion. We have accordingly revised the title to incorporate both technologies, as follows :

“Comparison and Validation of Rapid Molecular Testing Methods for Theranostic EGFR Alterations in Lung Cancer: Idylla versus ddPCR”

-Like de19, which shows the position of gene deletion in exon 19. State the location of the l858r mutation in exon 21.  

> we have clarified this information in the abstract:

L22 : “Targeting EGFR alterations, particularly the L858R (Exon 21) mutation and Exon 19 deletion (del19),…”

> and conserved this mention in the text :

L52 : “), or in exon 21, leading to the substitution of lysine at position 858 with arginine (L858R)…”

-In the abstract, the sum total of the mentioned samples is 121, while the set of purified DNAs is 122.  

> We apologize for the omission of the L861Q mutation. We have amended the abstract as follows:

L28 : “Additional L858R (N=24), L861Q (N=1), del19 (N=63), and WT samples (N=34)”

So that the total is really 122 (24 L858R, 1 L861Q, 63 del19, 34 WT)

-How were the 34 wild type negative samples confirmed not to carry any of the mutants?

>  We conducted Sanger and NGS AmpliSeq Ion Torrent analyses on both macro- and micro-dissected sections, both targeting EGFR exons 18, 19, 20, and 21. We have revised the text to provide this specific information as follows

L535 : “mutation status was determined by Sanger sequencing of exons 18-19-20 and 21, and after March 2016 by Next Generation Sequencing (NGS) with a custom amplicon panel (BED of 22 kb, covering EGFR exons 2-7-9-18-19-20 and 21…”

-A table should be added and positive samples should be shown based on the type and number of their mutations.  

> We agree, this data is indeed noteworthy. We have incorporated the results of sequencing for exon 18 to 21 (when available) into Table S2

-Statistical calculations in the calculation of sensitivity and specificity in the diagnosis of patients should be well described. The ROC curve should be  displayed.  

> We have clarified this question within the Materials and Methods section and included the ROC curve for ddPCR in the supplementary data (Figure S5). It is important to note that we were unable to construct an ROC curve for Idylla, as the Cq values for negative samples are unavailable

L354 : “As VAF is a continuous variable available for negative and positive samples (whereas DCq was not available for negative sample), the ROC curve was calculated with an AUC of 0.97 (Figure S5).”

L601: “… the ROC curve was obtained using the ROCR package. The Sensitivity was calculated as follow: Se=P/TP, representing the number of samples detected as mutated (P) among the true positive samples (determined by NGS/sanger). The specificity was calculated as follow : Spe= N/TN, meaning the number of samples detected as unmutated among the true negative samples (determined by NGS/Sanger).

L 643: “Figure S5 : ROC curve analysed for VAF values obtained by ddPCR.” 

-The comparison of the two Idylla and ddPCR methods in terms of the time required to perform the test and its cost should be added in the article.

We have evaluated the costs and modified the text as follows :

L195 : Table 1 : price comparison has been summarized

L 515 : “In our assessment, we found that the cost of Idylla was approximately twice that of ddPCR when starting from extracted DNA (see Table 1 and material and method section).”

L608 : in the material and methods section :

“4.4. Cost and Hands-on Time

A cost analysis was conducted for both single-patient and multi-patient scenarios, considering the expenses associated with hands-on time and reagent kit costs. Notably, this calculation excludes the costs of consumable materials not included in the kit, equipment procurement and maintenance. The hourly rate for technical labor was set at €41/hour, while biological interpretation was valued at €97/hour, following French regulations. For the Idylla system, the estimated technical time required was 10 minutes, encompassing DNA deposition and cartridge preparation, as well as the initiation of the analysis on the Idylla instrument. Biological interpretation took an additional 15 minutes for visualization and curve validation. The cartridge cost was €220. For ddPCR, the total cost was determined based on 1.4 hours of hands-on technical work, which included PCR mix preparation and droplet generation (45 minutes), droplet reading (20 minutes), and droplet selection for genotyping using Quantasoft software (20 minutes). Additionally, 20 minutes of pathologist time was considered, and reagent costs ranged from €60 to €24 for 1 to 10 simultaneous analyses (regardless of the number of samples, both negative and positive controls were required for each run). Importantly, the ddPCR experiment allowed for the management of several samples within 5 hours of hands-on time, whereas the Idylla device could process only one sample in 3 hours.”

Comments on the Quality of English Language

The manuscript has been corrected by the English proofreading department of the Hospices Civils de Lyon

We added this mention in the acknowledgment section

L 668 : “We thank Véréna Landel (Direction de la Recherche en Santé, Hospices Civils de Lyon) for her help in manuscript preparation.”